# ETV2 regulates PARP-1 binding protein to induce ER stress–mediated death in tuberin-deficient cells

Shikshya Shrestha[1] , Anthony Lamattina[1], Gustavo Pacheco-Rodriguez[2], Julie Ng[1], Xiaoli Liu[1], Abhijeet Sonawane[3], Jewel Imani[1], Weiliang Qiu[4] , Kosmas Kosmas[5] , Pierce Louis[1] , Anne Hentschel[1], Wendy K Steagall[2], Rieko Onishi[2], Helen Christou[5], Elizabeth P Henske[1], Kimberly Glass[4], Mark A Perrella[1,5] , Joel Moss[2], Kelan Tantisira[4,6], Souheil El-Chemaly[1]

**Lymphangioleiomyomatosis (LAM) is a rare progressive disease, characterized by mutations in the tuberous sclerosis complex genes (*TSC1* or *TSC2*) and hyperactivation of mechanistic target of rapamycin complex 1 (mTORC1). Here, we report that E26 transformation–specific (ETS) variant transcription factor 2 (ETV2) is a critical regulator of *Tsc2*-deficient cell survival. ETV2 nuclear localization in *Tsc2*-deficient cells is mTORC1-independent and is enhanced by spleen tyrosine kinase (Syk) inhibition. In the nucleus, ETV2 transcriptionally regulates poly(ADP-ribose) polymerase 1 binding protein (PARPBP) mRNA and protein expression, partially reversing the observed down-regulation of PARPBP expression induced by mTORC1 blockade during treatment with both Syk and mTORC1 inhibitors. In addition, silencing Etv2 or *Parpbp* in *Tsc2*-deficient cells induced ER stress and increased cell death in vitro and in vivo. We also found *ETV2* expression in human cells with loss of heterozygosity for *TSC2*, lending support to the translational relevance of our findings. In conclusion, we report a novel ETV2 signaling axis unique to Syk inhibition that promotes a cytocidal response in *Tsc2*-deficient cells and therefore maybe a potential alternative therapeutic target in LAM.**

## Introduction

Lymphangioleiomyomatosis (LAM) is a rare, multisystem disease associated with smooth muscle–like "LAM cells" over-proliferating in lungs, kidneys, and lymphatics. LAM cells have inactivating mutations in either tuberous sclerosis complex 1 or 2 (*TSC1* or *TSC2*) genes, leading to hyperactivation of the mechanistic/mammalian

target of rapamycin complex 1 (mTORC1) signaling (Carsillo et al, 2000; Cheadle et al, 2000; Goncharova et al, 2002; Crino et al, 2006; Johnson et al, 2016; Rosset et al, 2017). A pivotal clinical trial has led to the approval of the allosteric mTORC1 inhibitor sirolimus (rapamycin) by the Food and Drug Administration for use in LAM (McCormack et al, 2011). However, because of the cytostatic nature, continuous treatment with rapamycin is required to maintain lung function stability (Yao et al, 2014). Therefore, there is a need for a better understanding of tuberin-deficient cell death mechanisms, which could potentially lead to novel therapies.

We have previously shown that there is increased expression and activation of spleen tyrosine kinase (Syk) in both *Tsc2*-deficient cells and LAM lung nodules (Cui et al, 2017). Similar to rapamycin treatment, R406, a Syk inhibitor (SykI) had antiproliferative effects in *Tsc2*-deficient cells in vitro and in vivo (Cui et al, 2017). Syk-dependent regulation of mTOR signaling has been previously documented in B-cell lymphoma (Leseux et al, 2006; Fruchon et al, 2012) and acute myeloid leukemia (Carnevale et al, 2013). We demonstrated that mTORC1 inhibition altered Syk expression and activity, suggesting a feedback loop between the two pathways in *Tsc2*-deficient cells (Cui et al, 2017). Here, we sought to investigate regulatory pathways of Syk inhibition that are independent of its cross-talk with mTORC1 signaling in *Tsc2*-deficient cells and potentially identify target(s) of SykI that can have cytotoxic effects on *Tsc2*-deficient cells.

We performed an expression array of Syk and mTORC1 inhibition to identify unique regulatory mechanism(s) between the two signaling pathways. We then used Passing Attributes between Networks for Data Assimilation (PANDA) (Glass et al, 2013) to reconstruct treatment-specific transcription factor regulatory networks in *Tsc2*-deficient cells. Using these networks, we identified E26 transformation–specific (ETS) variant transcription factor 2 (ETV2) that targets a unique set of genes in

---

[1]Division of Pulmonary and Critical Care Medicine, Brigham and Women's Hospital, Harvard Medical School, Boston, MA, USA   [2]Division of Intramural Research, Pulmonary Branch, National Heart, Lung and Blood Institute (NHLBI), National Institutes of Health (NIH), Bethesda, MD, USA   [3]Department of Cardiovascular Medicine, Brigham and Women's Hospital, Harvard Medical School Boston, MA, USA   [4]Channing Division of Network Medicine, Brigham and Women's Hospital, Harvard Medical School, Boston, MA, USA   [5]Department of Pediatric Newborn Medicine, Brigham and Women's Hospital, Harvard Medical School, Boston, MA, USA   [6]Division of Pediatric Pulmonary and Critical Care Medicine, University of California San Diego, La Jolla, CA, USA

Correspondence: Sel-chemaly@bwh.harvard.edu; sshrestha4@partners.org

the SykI network independent of mTORC1 inhibition and is a critical regulator of *Tsc2*-deficient cell survival.

ETV2 is a member of the ETS family of transcription factors, which plays a key role in the development of hematopoietic and endothelial lineages (Liu et al, 2015; Garry, 2016). *Etv2* deficiency is associated with a complete blockage in blood and endothelial cell formation (Garry, 2016; Oliver & Srinivasan, 2010). In zebrafish, direct binding of ETV2 to the *Lyve1* and *Vegfr-3* promoter/enhancer results in the transcriptional regulation of those lymphatic markers (Davis et al, 2018). The role of ETV2 in lymphangiogenesis potentially has functional implications in LAM as there are lymphatic manifestations in LAM, including increased expression of VEGF-D in the serum of LAM patients (Seyama et al, 2006; Glasgow et al, 2008; Young et al, 2010). Studies also have demonstrated the potential lymphatic origin of LAM or *Tsc2*-deficient cells (Davis et al, 2013; Yue et al, 2016). Recently, studies have demonstrated ETV2 mRNA expression and amplification in various tumor specimens, including glioblastoma and adrenocortical carcinoma (Li et al, 2018; Zhao et al, 2018). The activation of ETV2 in tumor-associated endothelial cells was shown to contribute to tumor angiogenesis (Kabir et al, 2018).

We demonstrated that ETV2 differentially regulates a unique set of genes in the SykI treatment network, including poly(ADP-ribose) polymerase 1 (PARP1) binding protein (PARPBP). PARPBP, also known as PARI or C12orf48, is a replisome-associated protein that plays important roles during replication stress and DNA repair through its interaction with PARP-1, PCNA, and RAD51 (Varisli, 2013; Burkovics et al, 2016; Nicolae et al, 2019). PARPBP overexpression is associated with hyperproliferation in pancreatic cancers, hepatocellular carcinoma, and myeloid leukemia (Piao et al, 2011; O'Connor et al, 2013; Nicolae et al, 2019; Yu et al, 2019). PARP-1 inhibitors are promising antineoplastic agents (Malyuchenko et al, 2015) and have been shown to selectively inhibit proliferation and induce apoptosis in *Tsc2*-deficient cells (Malyuchenko et al, 2015). However, the role of PARPBP in LAM or *Tsc2*-deficient cells and its regulation are largely unknown. Our data showed that ETV2 transcriptionally regulates PARPBP. We further observed increased ER stress and increased *Tsc2*-deficient cell death when *Etv2* or *Parpbp* was silenced. Hence, targeting ETV2, with its potential for cytocidal cellular responses in *Tsc2*-deficient cells, might offer a therapeutic advantage in LAM over rapalogs alone, which primarily act as cytostatic drugs.

# Results

## Gene expression profiling in rapamycin- versus SykI-treated *Tsc2*-deficient cells

To discern the effects on gene expression of mTORC1 and Syk inhibition in *Tsc2*-deficient cells, we ran Rat Gene 2.0 ST microarrays with ELT3-V cells treated with DMSO, SykI, or rapamycin for 24 h. A heatmap generated using the top 485 most differentially expressed genes in each treatment condition compared with the control (DMSO) showed four distinct clusters (Fig 1A). We validated the gene expression data for the top 8 differentially expressed genes from each cluster using RT-qPCR (Fig S1A–D). The principal component analysis plot, based on PC1, revealed DMSO-treated replicates

clustered very distinctly from both SykI- and rapamycin-treated replicates, suggesting considerable changes in gene expression in the treatment groups compared with DMSO (Fig 1B). In contrast, only a minute separation in the clusters was observed between the two treatment groups, suggesting largely concordant changes in gene expression (Fig 1B).

The top cluster (Cluster 1, red) in the heatmap included 201 genes that were similarly down-regulated in both SykI- and rapamycin-treated cells compared with DMSO-treated cells (Fig 1A). Gene ontology (GO) enrichment analysis on these genes showed up-regulation in biological processes that include oxidation–reduction and L-serine biosynthesis (Fig 1C). The second cluster (Cluster 2, green) included 107 genes that were down-regulated in both treatment groups compared with DMSO but demonstrated greater down-regulation with rapamycin than with SykI (Fig 1A). These genes corresponded to 17 significantly up-regulated biological processes, most of which were associated with the cell cycle (Fig 1C). Only one gene (*Mospd1*) was down-regulated by SykI treatment, but not rapamycin, comprising the third cluster (Cluster 3, blue). RT-qPCR validation however showed down-regulation of *Mospd* in both SykI and rapamycin treatment groups compared with DMSO (Fig S1C). The fourth cluster (Cluster 4, pink) included 176 genes that were up-regulated in both treatment groups compared with DMSO and corresponded to a significant up-regulation of biological processes associated with cellular migration (Fig 1A and C).

Overall, differential expression data from the microarray indicated that SykI and rapamycin treatments similarly impacted gene expression in *Tsc2*-deficient cells. A subset of genes, comprising Cluster 2, were observed to be more down-regulated with mTORC1 inhibition than with Syk inhibition.

## Network analysis reveals putative SykI-specific regulation driven by the transcription factor Etv2

To further understand potential gene regulation differences between SykI and rapamycin treatments, we used PANDA analysis to integrate information from treatment-specific gene expression and transcription factor binding motifs and construct transcriptional regulatory networks for the three treatment groups using pairwise comparisons between two groups at a time: SykI versus DMSO, rapamycin versus DMSO, and SykI versus rapamycin. Edge weight differences for the top 10,000 (TF, gene) pairs with the largest absolute differences of edge weights between two networks SykI and rapamycin were plotted (Fig 1D). Consistent with the finding from differential gene expression and pathway analyses, a high correlation between the edge weights for the two treatment groups was observed; however, we also found regulatory edges that were more strongly supported by either SykI (red) or rapamycin (blue) treatment. Based on the top 10,000 (TF, gene) pairs, SykI and rapamycin networks consisted of 54 unique TFs in either of the two networks, and most TFs belonged to the members of the E26 transformation-specific (Ets) family of TFs (Table 1). *Ets* variant 2 (ETV2) was among the top 20 TFs regulating the highest numbers of genes in each network. Because evidence suggested a potential lymphatic endothelial origin of LAM cells (Davis et al, 2013; Yue et al, 2016) and that ETV2 is critical for lymphangiogenesis (Davis et al, 2018), we further investigated the role of ETV2.

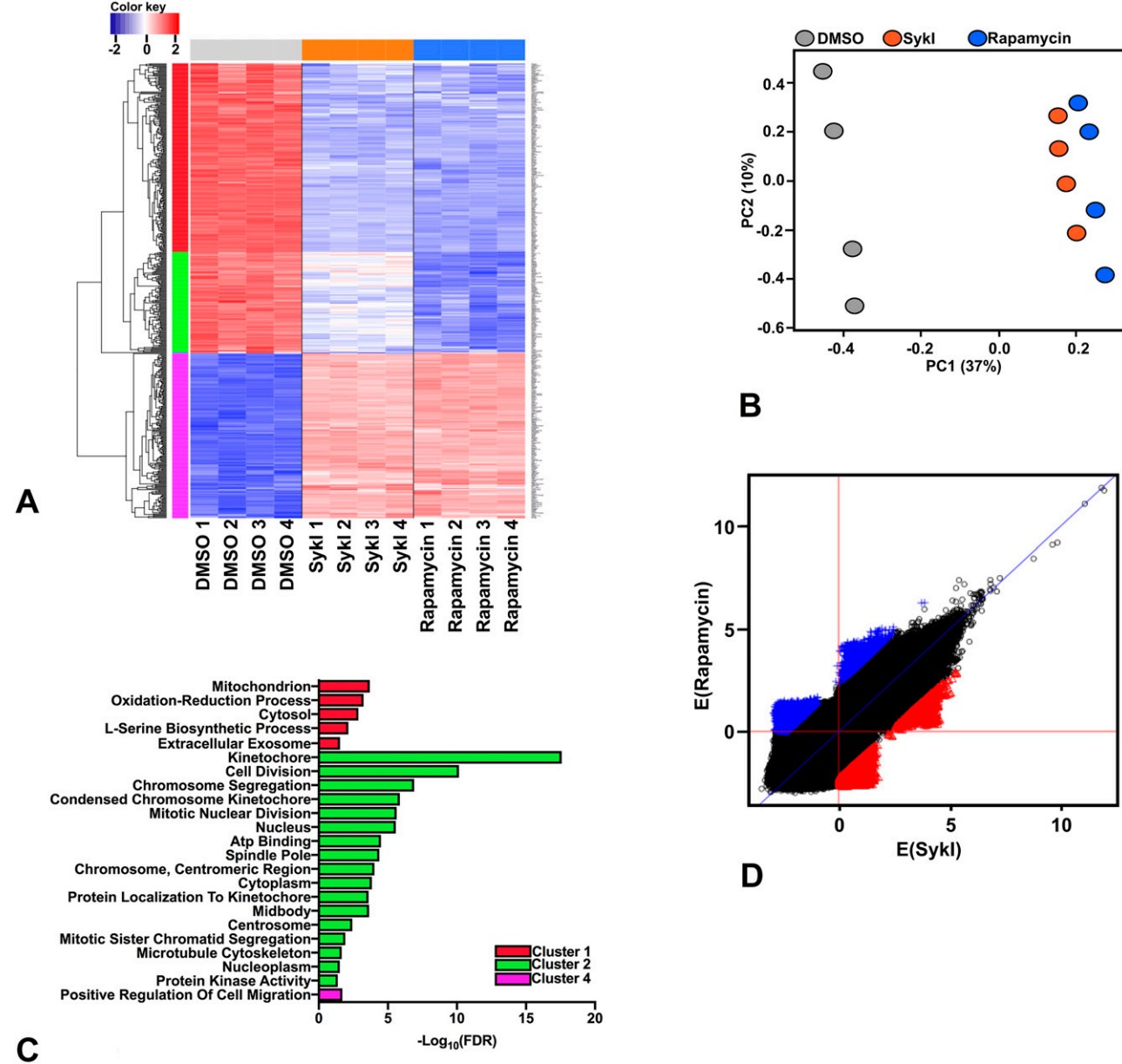

**Figure 1.  Gene expression profiling in rapamycin- versus SykI-treated ELT3-V cells.**
**(A)** *Tsc2*-deficient Eker rat (ELT3-V) cells were treated with SykI (R406; 1 µM), rapamycin (20 nM), or DMSO for 24 h. Data were obtained using Affymetrix Rat Gene 2.0 ST GeneChip. The diagram represents 485 differentially expressed probes (q < 0.01) with at least twofold change between conditions. Each column represents a single experiment and each row a single gene. The colors are scaled so that red and blue indicate z-scores of ≥2 or ≤−2, respectively, and white indicates a z-score of 0 (row-wise mean). Color blocks on top, gray, orange, and blue represent different treatment groups, DMSO, SykI, and rapamycin, respectively. Color blocks on the left, red, green, blue, and pink represent four different clusters, Clusters 1–4. **(B)** Principal component analysis of samples treated with DMSO, rapamycin, or SykI for 24 h. Principal component analysis was performed across all probes on the microarray and a bi-plot of PC1 against PC2 was plotted to show clustering of different treatment groups and condition replicates. **(A, C)** Gene ontology enrichment analysis for significantly up-regulated biological processes using genes in each cluster in the heatmap (A). Gene ontology terms for Clusters 1, 2, and 4 are presented in red, green, and pink, respectively. **(D)** Scatter plot for the SykI and rapamycin networks predicted by PANDA for the top 10,000 edges (TF, gene) in each of the networks models, rapamycin treatment, and SykI treatment models. Each point in the graph represents a difference in edge weight connecting a transcription factor to a target gene between two networks. Red points represent the 10,000 (TF, gene) pairs having largest weight in SykI network, and blue points represent (TF, gene) edges having the largest weight in rapamycin network.

## SykI, but not rapamycin, treatment induced ETV2 translocation into cell nuclei

To understand a potential role for ETV2 in mediating SykI-dependent function in *Tsc2*-deficient cells, we investigated *Etv2* gene and ETV2 protein expression in *Tsc2*-deficient ELT3-V cells treated with DMSO, SykI, or rapamycin for 24 h. No significant differences in both total mRNA and total protein expression were observed with either SykI or rapamycin treatment compared with DMSO (Fig 2A–C). SYK has previously been shown to regulate the

**Table 1. TF with highest nDiff in SykI and Rapa networks identified using top 10,000 (TF, genes) pairs.**

| Gene | TF motif ID | TF family | nSykI | nRapa | nDiff | nOverlap | nRatio |
|------|-------------|-----------|-------|-------|-------|----------|--------|
| Elk4 | M0703 | ETS domain | 1,106 | 1,064 | 42 | 0 | 1.04 |
| Elk3 | M0692 | ETS domain | 1,051 | 1,005 | 46 | 0 | 1.05 |
| Gm5454 | M0710 | ETS domain | 1,039 | 972 | 67 | 0 | 1.07 |
| Etv3 | M0689 | ETS domain | 916 | 862 | 54 | 0 | 1.06 |
| Erfl | M0714 | ETS domain | 912 | 869 | 43 | 0 | 1.05 |
| Elf4 | M0706 | ETS domain | 687 | 666 | 21 | 0 | 1.03 |
| Etv6 | M0705 | ETS domain | 665 | 618 | 47 | 0 | 1.08 |
| Elk1 | M6207 | ETS domain | 621 | 590 | 31 | 0 | 1.05 |
| Id4 | M5571 | Basic helix–loop–helix | 346 | 356 | −10 | 0 | 0.97 |
| Erf | M5398 | ETS domain | 327 | 327 | 0 | 0 | 1.00 |
| Myod1 | M2299 | Basic helix–loop–helix | 297 | 308 | −11 | 0 | 0.96 |
| Tcfl3 | M0184 | Basic helix–loop–helix | 241 | 242 | −1 | 0 | 1.00 |
| Myog | M2300 | Basic helix–loop–helix | 200 | 201 | −1 | 0 | 1.00 |
| Ehf | M0696 | ETS domain | 190 | 202 | −12 | 0 | 0.94 |
| Tcf12 | M2317 | Basic helix–loop–helix | 160 | 170 | −10 | 0 | 0.94 |
| Gabpa | M4568 | ETS domain | 117 | 134 | −17 | 0 | 0.87 |
| **Etv2** | **M5421** | **ETS domain** | **116** | **117** | **−1** | **0** | **0.99** |
| Zfp740 | M0429 | C2H2–zinc finger | 98 | 120 | −22 | 0 | 0.82 |
| Mesp1 | M5627 | Basic helix–loop–helix | 94 | 111 | −17 | 0 | 0.85 |
| Tal1 | M6358 | Basic helix–loop–helix | 93 | 110 | −17 | 0 | 0.85 |

nSykI, genes connected by given TF in SykI network; nRapa, genes connected by given TF in Rapa network; nDiff, nSykI–nRapa; nOverlap, Genes overlapping between SykI and Rapa; nRatio, nSykI/nRapa. *ETV2* is among the TF with highest nDiff and is presented as bold entry.

nuclear translocation of transcription factors, including nuclear factor (erythroid-derived 2)-like 2 (Nrf2) and Ikaros (Uckun et al, 2012; Park et al, 2018). Therefore, we also examined the effect of SykI and rapamycin treatment on ETV2 localization. Western blot analysis of ELT3-V cell fractions showed significantly increased nuclear ETV2 protein levels in SykI-treated cells compared with DMSO-treated cells (Fig 2D and E), and importantly, treatment with rapamycin did not result in ETV2 nuclear accumulation. To further confirm the specificity of our findings, we silenced SYK in ELT3-V cells and similarly observed ETV2 nuclear accumulation (Fig S2A–G). These data collectively suggest that Syk inhibition, but not mTORC1 inhibition, drives ETV2 nuclear translocation.

### *Etv2* silencing induces ER stress and leads to *Tsc2*-deficient cell death

In addition to its role in endothelial and lymphatic lineage, ETV2's role as an important regulator of endothelial cell survival, cell cycle, and proliferation has been previously studied (Abedin et al, 2014; Singh et al, 2019). To assess the effect of ETV2 alteration on *Tsc2*-deficient cells, we silenced Etv2 in ELT3-V cells (Fig 3A–C). Silencing *Etv2* led to a significant increase in Annexin V–positive cells, indicating increased cell apoptosis/death with reduced ETV2 expression (Fig 3D and E). We then sought to determine whether the *Etv2* silencing–mediated cell death is related to ER stress as *Tsc2*-

deficient cells are known to be sensitive to ER stress (Ozcan et al, 2008; Kang et al, 2011). As shown in Fig 3, the expression of ER stress markers CHOP (Fig 3F and G) and phosphorylated (p)EIF-2α (Fig 3F and H) were markedly elevated in *Etv2*-silenced ELT3-V cells, suggesting increased ER stress. Importantly, the ER stress response was associated with increased cell death as demonstrated by the increase in cleaved PARP (cPARP) expression (Fig 3F and I).

Contrarily, the cell death response was not observed when *Etv2* was silenced in tuberin-reexpressing ELT3-T cells (Fig S3A–D). Moreover, silencing *Etv2* induced cell death in *Tsc2*-deficient mouse embryonic fibroblasts, suggesting that the effect of *Etv2* silencing is not cell type-specific (Fig S4A–D). These data strongly support the critical importance of ETV2, specifically in mTOR-hyperactivated cell survival.

To further confirm the ETV2-mediated induction of ER stress, we examined the role of ETV2 in the regulation of stress granule (SG) dynamics. We evaluated oxidant-induced SG formation in ELT3-V cells in which *Etv2* was silenced by siRNA. ELT3-V cells transfected with Etv2 or Scr siRNA were exposed to arsenite (0.5 mM) for 40 min, and SG formation was quantified by immunofluorescence using the marker G3BP1 (Anderson & Kedersha, 2002; Kosmas et al, 2021). ELT3-V cells transfected with Etv2 siRNA had a significantly increased number of SGs after arsenite treatment compared with cells transfected with Scr siRNA (1.5-fold increase; Fig 3J–K). These data suggest that ETV2 plays a role in SG assembly in response to oxidant stress.

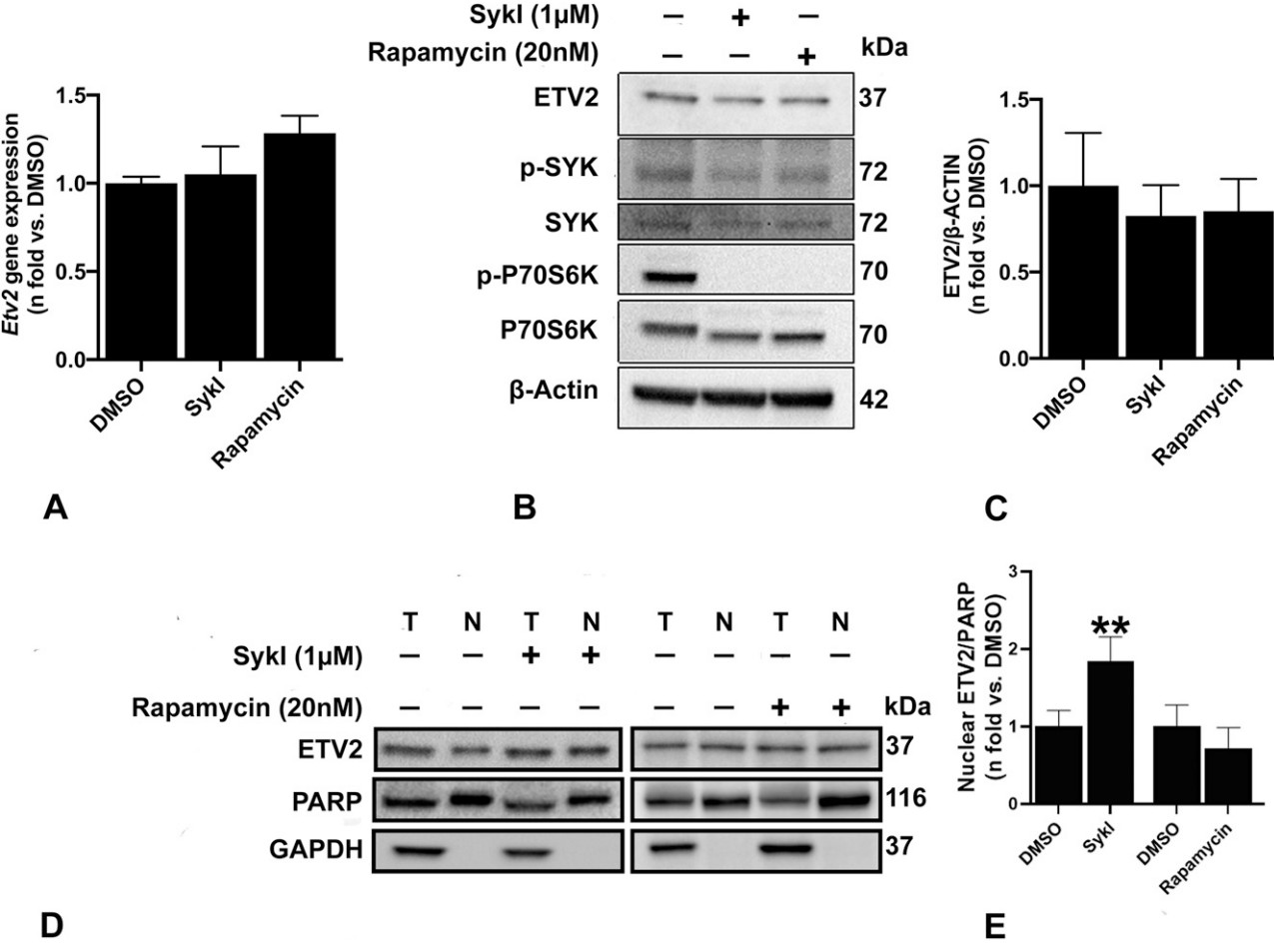

**Figure 2. Effect of SykI and rapamycin treatment on ETV2 expression and localization.**
**(A)** Real-time qPCR analysis of rat *Etv2* mRNA in *Tsc2*-deficient ELT3-V cells treated with SykI or rapamycin for 24 h compared with DMSO-treated cells. Data represent mean ± SEM of three independent experiments. **(B, C)** Immunoblot analysis of equal amount of rat ETV2 protein from ELT3-V cells treated with DMSO, SykI, or rapamycin for 24 h. Antibodies against rat ETV2, p-SYK, SYK, p-P70S6 kinase, and P70S6 kinas*e* were used, and *β*-actin was used as a loading control. Band intensities of ETV2 were assessed, and ratios of ETV2/*β*-actin were calculated for each treatment group. Results were expressed relative to DMSO. Data are means ± SEM of at least three independent experiments (ETV2/*β*-actin: *P* > 0.05; *t* test). **(D, E)** ELT3-V cells were treated with DMSO, SykI, or rapamycin as in (B). Samples from equal fractions of total (T) and nuclear (N) protein lysates were separated by SDS–PAGE and transferred to polyvinylidene difluoride membranes, which were incubated with antibodies against ETV2, PARP (nuclear marker), and GAPDH (cytoplasmic marker). Band intensities of ETV2 were analyzed, and ratios of ETV2/PARP and ETV2/GAPDH were calculated for each group. Results were expressed relative to DMSO. Data are means ± SEM of at least three independent experiments (ETV2/PARP: *P* < 0.05, **P* < 0.01; *t* test). Source data are available for this figure.

### Syk inhibition regulates *Parpbp* expression

An examination of the PANDA analysis showed that ETV2 regulated 116 unique genes in the SykI network and 117 unique genes in the rapamycin network. Parp1 binding protein (PARPBP) is an important component of DNA replication and damage response pathways and is differentially expressed in various cancers (Varisli, 2013; Feng et al, 2014; Uhlen et al, 2015). Based on our microarray data and PANDA analysis, we found that Parpbp was regulated by *Etv2* uniquely under SykI treatment and demonstrated the highest fold change between treatments compared with other ETV2-regulated genes within that network (Table 2).

RT-qPCR analysis showed that both SykI and rapamycin treatments significantly reduced *Parpbp* expression compared with DMSO; however, the reduction was of significantly greater magnitude in the rapamycin treatment group than in SykI (Fig S5A). We hypothesized that the relatively higher levels of *Parpbp* mRNA with SykI treatment than with rapamycin treatment are due to ETV2 nuclear translocation. Accordingly, silencing of ETV2 significantly reduced overall *Parpbp* expression (Fig S5B), and abrogated the difference in the *Parpbp* expression level between the two treatment groups (Fig 4A), thus confirming the important role for ETV2 in Parpbp regulation in *Tsc2*-deficient cells.

Genome-wide analysis has revealed that ETV2 is an ETS factor with various transcriptional targets containing the consensus sequence of 5′-CCGGAA/T-3′ and a core GGA/T motif (Wei et al, 2010; Lee et al, 2019). To examine whether ETV2 transcriptionally regulates *Parpbp*, *Parpbp* promoter fragments with one or no ETV2 consensus binding sequence were cloned into pGL3_Basic luciferase vector and transfected separately into ELT3-V cells with or without *Etv2*

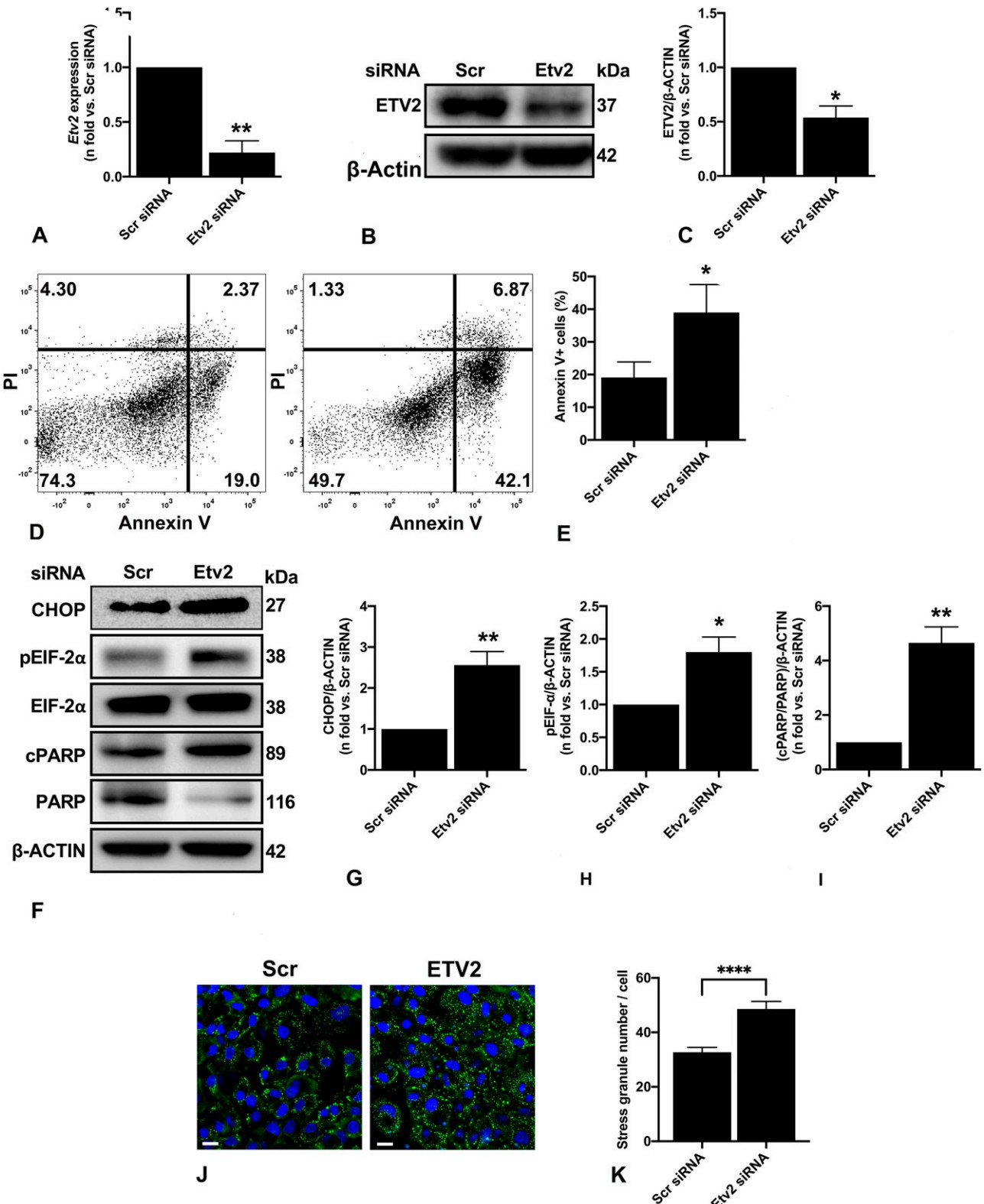

**Figure 3. Silencing *Etv2* induces ER stress and cell death in *Tsc2*-deficient cells.**

*Tsc2*-deficient cells (ELT3-V) were transfected with either Scr or Etv2 siRNA for 48 h. **(A)** Real-time qPCR analysis of rat *Etv2* mRNA. The histogram represents fold change in mRNA expression compared with the Scr control. Data are means ± SEM of at least three independent experiments (**$P < 0.01$; *t* test). **(B, C)** Equal amounts of protein from whole-cell lysates were analyzed by Western blotting with antibodies against ETV2 and β-actin. The ratio of ETV2 to β-actin density was expressed as the fold-change relative to Scr siRNA. Data are means ± SEM of at least three independent experiments (*$P < 0.05$; *t* test). **(D, E)** PI and FITC AnnexinV apoptosis assay and flow cytometry

silencing to determine promoter activity. The ELT3-V cells without *Etv2* silencing transfected with pGL3_promoter-386 construct (i.e., the −311- to +75-bp putative promoter region) with one ETV2 consensus binding site displayed significantly higher luciferase activity than cells transfected with the pGL3_promoter-150 construct (i.e., the −75- to +75-bp putative promoter region) with no ETV2 consensus binding site (Fig 4B). In addition, silencing of *Etv2* significantly reduced the luciferase activity of the pGL3_promoter-386 construct but showed no effect in the pGL3_promoter-150 construct. These results demonstrated the role of ETV2 in *Parpbp* transcriptional regulation.

### *Parpbp* silencing leads to ER stress in *Tsc2*-deficient cells

*Parpbp* silencing has been implicated in increased apoptosis and ER stress in myeloid leukemia cells (Nicolae et al, 2019). We hypothesized that ETV2-dependent ER stress and increased cell death in *Tsc2*-deficient cells are mediated by changes in PARPBP expression. To specifically investigate the role of PARPBP in the induction of ER stress and cell death in *Tsc2*-deficient cells, we silenced *Parpbp* in ELT3-V cells with Parpbp siRNA. Parpbp siRNA resulted in ~80% and 50% reduction in *Parpbp* mRNA and PARPBP protein, respectively, compared with Scr siRNA (Fig 4C–E). Consequently, there was a significant increase in ER stress markers, including CHOP and pEIF-2α (Fig 4F–H), and a significant increase in cell death marker, cPARP (Fig 4I). These observations support our hypothesis that the regulation of *Parpbp* expression by ETV2 contributes to ER stress and increased cell death in *Tsc2*-deficient cells following *Etv2* silencing.

### *Etv2* silencing leads to *Tsc2*-deficient cell death in vivo

To determine the effect of *Etv2* silencing on *Tsc2*-deficient cell survival in vivo, ELT3-V-luciferase cells were transiently transfected with Scr or Etv2 siRNA for 24 h. Cells were intravenously injected into female SCID mice, and the level of bioluminescence was evaluated in the lungs 4-h postinjection. Similar levels of bioluminescence intensity were observed in mice injected with both Scr siRNA– and Etv2 siRNA–transfected cells (Fig 5A and B). By 24-h and 48-h postinjection, there was a decrease in bioluminescence in both groups; however, the bioluminescence intensities in the mice injected with *Etv2*-silenced cells were significantly lower than those in mice injected with Scr siRNA–transfected cells (Fig 5A and B). We then measured rat-specific DNA in cDNA samples obtained from lung tissue and peripheral blood using RT-qPCR. The levels of rat-specific DNA in both lung tissue and peripheral blood were significantly lower in the mice injected with *Etv2*-silenced cells than in the Scr control (Fig 5C and D). Consistent with our in vitro findings,

these data suggest that *Etv2* silencing leads to *Tsc2*-deficient cell death in vivo.

### *ETV2* is expressed in human LAM samples

Analysis of single-cell RNA-seq data from three LAM lung samples using Seurat identified seven different unique clusters of cells (Fig S6A), including alveolar type II (AT2) cells, conventional dendritic cells (cDC), endothelial cells, fibroblasts, macrophages, natural killer cells, and LAMCORE (Guo et al, 2020). A total of 121 LAMCORE cells defined as those which expressed (>0) LAM markers, *PMEL*, *ACTA2*, and *FIGF* (Guo et al, 2020) were identified (Fig 6A). One hundred LAMCORE cells were clustered distinctly, 13 were clustered within fibroblast population, and nine LAMCORE cells were distributed within other cell populations. We also examined ETV2 expression (>0) in LAMCORE cells and demonstrated that 96% of LAMCORE cells were positive for ETV2 expression (Fig 6A). Other cell types also expressed *ETV2* however LAMCORE cells were among the highest expressing cells for in the LAM lung (Fig S6A).

Likewise, sorted cultured cells expressing *CD44* and *CD44v6* (known to have a loss of heterozygosity [LOH] for TSC2) isolated from five different explanted lung samples (Samples 1–5, Fig 6B), and cells expressing *CD235a* isolated from eight blood samples from patients undergoing lung transplantation (Samples 6–13, Fig 6B) were analyzed by RT-PCR for *ETV2* expression. Primers designed for PCR targeted the amplification of a 106-bp *ETV2* mRNA sequence. The PCR product of the target region from each sample was visualized by electrophoresis on 2% agarose gels. All samples yielded a band at 106 bp, demonstrating the expression of *ETV2* in all human LAM samples (primer only experimental control yielding no band, indicated by "Blank" is displayed in Fig S6B).

## Discussion

We have previously shown that similar to mTORC1 inhibition, Syk inhibition leads to decreased proliferation of tuberin-deficient cells in vitro and in vivo (Cui et al, 2017). In this study, we used gene profiling and network-based approaches with PANDA analyses and identified ETV2 as a regulatory transcription factor uniquely altered under Syk inhibition and not under mTORC1 inhibitory conditions.

ETV2 is a well-known regulator of blood and endothelial cell lineages during development (Li & Sidell, 2005; Garry, 2016). To date, the role of ETV2 in *Tsc2*-deficient tumors has not been investigated. Our data demonstrated that although Syk inhibition does not affect the expression of ETV2 at mRNA or protein levels, it prompted the translocation of ETV2 into *Tsc2*-deficient cell nuclei. Importantly,

were used to measure cell apoptosis induced by ETV2 silencing in ELT3-V cells. Representative flow cytometry plots for Scr and Etv2 siRNA are presented. The total percent of Annexin V positive (+) cells in each group was quantified from three independent experiments (*$P < 0.05$; $t$ test). **(F, G, H, I)** Equal amounts of proteins were separated by electrophoresis and transferred to polyvinylidene difluoride membranes, which were reacted with antibodies against CHOP, pEIF-2α, total EIF-2α, cleaved cPARP, and uncleaved PARP. β-Actin was used as a loading control. **(G, H, I)** Ratios of CHOP (G), pEIF-2α/EIF-2α (H), and cPARP/PARP (I) to β-actin were expressed as fold change to Scr siRNA. Data represent means ± SEM of at least three independent experiments (*$P < 0.05$, **$P < 0.01$; $t$ test). **(J, K)** ELT3-V cells were treated with 0.5 mM NaAsO$_2$ (arsenite) for 40 min. Cells were stained for G3BP1 (green) to detect stress granules, and DAPI (blue) was used to visualize the nuclei. A total of 221 cells per siRNA group were imaged and quantified with CellProfiler. Scale bar, 20 μm, ****$P < 0.0001$.
Source data are available for this figure.

**Table 2. List of top 10 genes identified with PANDA analysis regulated by *Etv2* included in the SykI-specific network.**

| Gene symbol | Gene name | Fold change | *P*-value | False discovery rate q-value |
|---|---|---|---|---|
| *Parpbp* | PARP1 binding protein | −1.38 | 0.00 | 0.05 |
| *Fam111a* | Family with sequence similarity 111, member A | −1.22 | 0.00 | 0.03 |
| *Ppp1r16b* | Protein phosphatase 1, regulatory subunit 16B | 1.22 | 0.03 | 0.38 |
| *Mtus1* | Microtubule-associated scaffold protein 1 | 1.16 | 0.01 | 0.29 |
| *Adarb2* | Adenosine deaminase, RNA-specific, B2 | 1.16 | 0.01 | 0.25 |
| *Gldn* | Gliomedin | −1.14 | 0.02 | 0.35 |
| *Rnf219* | Ring finger protein 219 | −1.14 | 0.03 | 0.37 |
| *Il20rb* | Interleukin 20 receptor subunit beta | −1.13 | 0.01 | 0.23 |
| *Vamp1* | Vesicle-associated membrane protein 1 | 1.11 | 0.04 | 0.41 |
| *Gprc6a* | G protein-coupled receptor, class C, group 6, member A | −1.10 | 0.03 | 0.40 |

rapamycin treatment does not affect the expression or nuclear localization of ETV2, suggesting that ETV2 may drive a potential mTORC1-independent transcriptional pathway in *Tsc2*-deficient cells. The presence of cAMP response element (CRE) sequences and therefore the regulation of ETV2 by PKA signaling have been shown previously (Yamamizu et al, 2012). Likewise, the interaction between SYK and PKA has also been described (Yu et al, 2013). Therefore, the nuclear translocation of ETV2 during SykI treatment could potentially be driven in a PKA-dependent manner. Further elucidation of this pathway will require additional investigation.

A known target of ETV2, acting as a transcription factor, are genes involved in the regulatory networks for hematopoietic and endothelial lineages, including Flk1 (Kim et al, 2019). We used PANDA network analysis and identified multiple genes that could be potential transcriptional targets of ETV2 following Syk inhibition in *Tsc2*-deficient cells. The most differentially regulated gene was *Parpbp*. Interaction between ETV1, EWS-ERG fusion genes, and EWS-FLI1 fusion genes, members of ETS family transcription factors, and PARP1, a PARPBP-interacting partner has been demonstrated in Ewing sarcoma and prostate cancers (Feng et al, 2014). Our study is the first to demonstrate that ETV2 transcriptionally regulates *Parpbp*. Both SykI and rapamycin result in mTORC1 blockade (Cui et al, 2017). Our data showed that treatment with both SykI and rapamycin resulted in decreased *Parpbp* levels, which we attribute to mTORC1 blockade. The rebound increase in *Parpbp* levels with Syk inhibition is likely due to ETV2 nuclear translocation, which is seen only with Syk inhibition and not mTORC1 inhibition.

More importantly, to investigate the functional role of ETV2 in *Tsc2*-deficient cells, we silenced *Etv2* and demonstrated that silencing of *Etv2* induced ER stress, leading to increased cell death both in vitro and in vivo. Our finding of ER stress–mediated *Tsc2*-deficient cell death extends prior knowledge on the contribution of ETV2 to cell survival and apoptosis (Abedin et al, 2014; Singh et al, 2019) as well as the susceptibility of *Tsc2*-deficient cells to ER stress (Ozcan et al, 2008; Kang et al, 2011). It was recently shown that TSC2 physically interacts with the SG protein G3BP1 and that SGs are increased in *Tsc2*-deficient cells (Kosmas et al, 2021). SGs may represent a mechanism to temporarily sequester transcripts during transient stress, including oxidant stress. Our data showed that silencing *Etv2* increased SGs in *Tsc2*-deficient cells in the setting of short-term arsenite-induced oxidant stress, suggesting that multiple mechanisms impact SGs in TSC. Furthermore, we also demonstrated that *Parpbp* silencing, similar to silencing of *Etv2*, resulted in increased ER stress and increased cell death in *Tsc2*-deficient cells. This suggested that ETV2 induces ER stress and increases cell death in *Tsc2*-deficient cells via *Parpbp* regulation (Fig 7). We also showed that MEFs deficient in tuberin are also sensitive to ETV2 depletion, resulting in cell death. Importantly, we also showed that cells expressing TSC2 are not susceptible to ETV2-mediated cell death.

ELT3 cells share some important common features with LAM cells including tuberin deficiency, activation of mTOR, and perhaps a common uterine origin (Guo et al, 2020). We elected to use ELT3 cells in an in vivo model of *Tsc2*-deficient cell colonization to the lung. At 4-h postinjection, there was no difference in bioluminescence intensity over the lungs between cells where *Etv2* was silenced and controls, suggesting that an identical number of cells successfully colonized the lungs. The decrease in bioluminescence over the subsequent time points in cells where *Etv2* was silenced strongly supports our hypothesis that ETV2 is critical for tuberin-deficient cell survival in vivo. Finally, we validated *ETV2* expression in human LAM cells in three different ways. First, we identified LAM cells in the recently published single-cell RNAseq data and found that these cells overwhelmingly expressed *ETV2*. Second, in a mixed cell population isolated from LAM lungs, we found that cells with LOH for *TSC2* expressed *ETV2*. Finally, LAM cells isolated from peripheral blood from female patients with LAM also expressed *ETV2*.

Taken together, the combination of the in vitro, in vivo, and human data strongly supports our hypothesis of a critical role for ETV2 in *Tsc2*-deficient cell survival (Fig 7A and B). Therefore, identifying molecules that target ETV2 expression or prevent its nuclear trafficking could potentially be of important therapeutic benefit in LAM and other tumors driven by mTORC1 activation.

# Materials and Methods

### Cell culture

*Tsc2*-deficient Eker rat uterine leiomyoma (ELT3-V) cells were cultured using DMEM containing 10% fetal bovine serum and

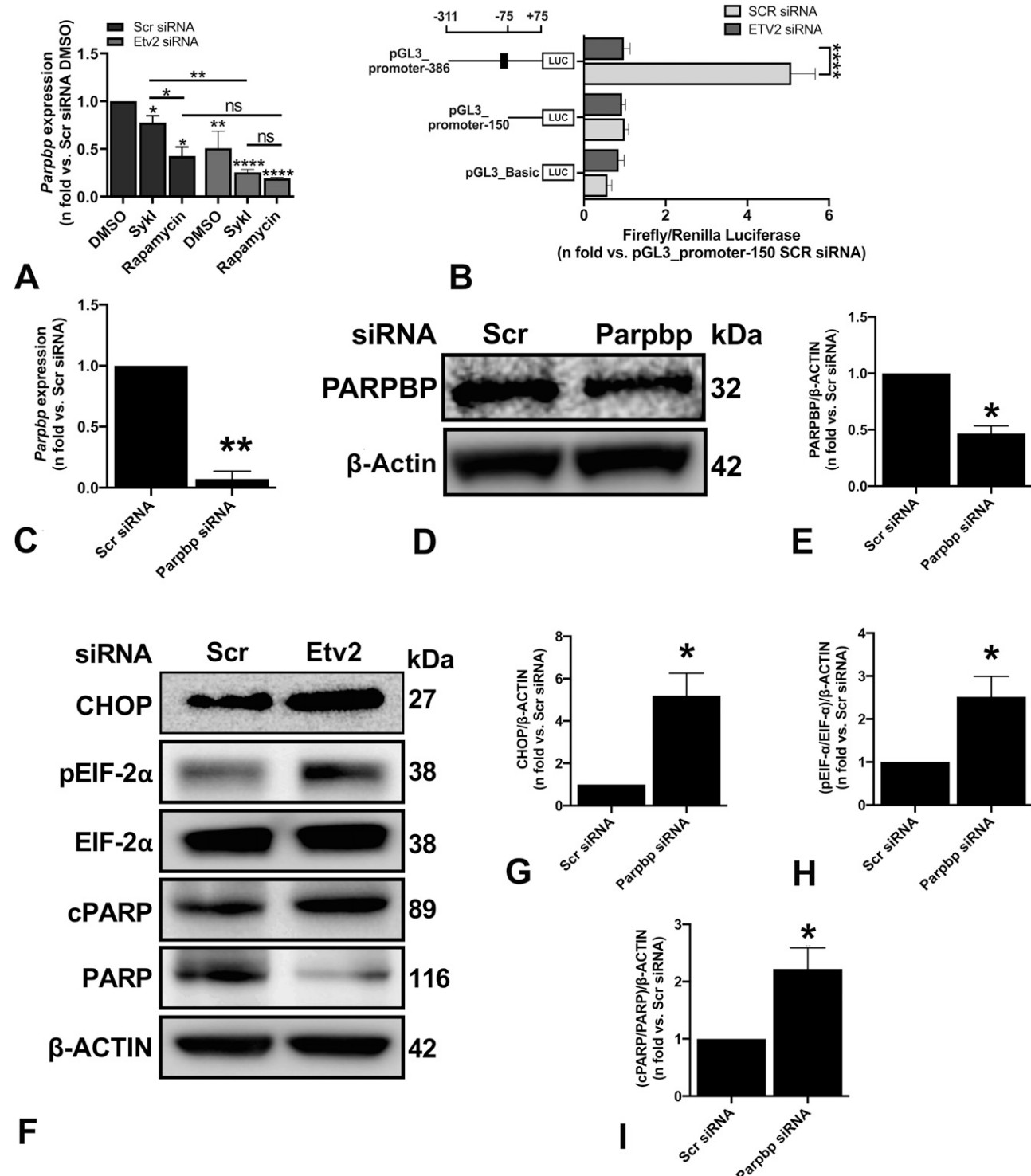

**Figure 4. *Etv2* silencing-induced changes in PARPBP lead to ER stress and cell death in *Tsc2*-deficient cells.**
**(A)** ELT3-V cells were transfected with Scr siRNA and Etv2 siRNA for 24 h and treated with DMSO, SykI, or rapamycin for an additional 24 h. RT–qPCR was carried out on an equal amount of total RNA (1 μg) converted to cDNA to analyze the transcript levels of *Parpbp* gene. The histogram represents fold change in mRNA expression relative to Scr siRNA with DMSO treatment. Data are means ± SEM of at least three independent experiments (*$P < 0.05$, **$P < 0.01$, ***$P < 0.001$; *t* test). **(B)** ELT3-V cells were transfected with Scr siRNA and Etv2 siRNA for 24 h, followed by transfection with internal control, pGL3_Renilla plasmid, and pGL3_Basic vector or *Parpbp* promoter constructs for an additional 24 h. Firefly luciferase and Renilla luciferase activities of negative control, pGL3_Basic vector and *Parpbp* promoter constructs with and without ETV2-binding site, pGL3_promoter-386, and pGL3_promoter-150 were measured. Each set of luciferase data is normalized to internal Renilla control. Data are presented relative to the pGL3_promoter-150 Scr siRNA sample. Data are means ± SEM of at least three independent experiments (****$P < 0.0001$; *t* test). **(C)** ELT3-V was transfected with either Scr or Parpbp siRNA for 48 h. Real-time qPCR analysis of rat *Parpbp* mRNA was performed. The histogram represents fold change in mRNA expression compared with the Scr

incubated at 37°C in a humidified 5% $CO_2$ atmosphere. For each experiment, cells were serum-starved overnight (16 h) prior to treatment with the vehicle control, DMSO (Sigma-Aldrich), Syk inhibitor (SykI) R406 (1 µM; Selleckchem), or rapamycin (20 nM; LC Laboratories).

## GeneChip hybridization and differential expression analyses

ELT3-V cells were treated with DMSO, rapamycin, or SykI for 24 h (n = 4 biological replicates per group). RNA was extracted using the RNeasy Mini Kit (QIAGEN). All subsequent sample preparation and GeneChip (Rat Gene 2.0 ST arrays) processing were performed at the Boston University Microarray and Sequencing Resource Core facility using an input of 1 µg of total RNA from each sample. Analysis was performed using the Bioconductor software suite (version 2.12) (Gentleman et al, 2004). Chip definition file (Dai et al, 2005) was processed using a robust multi-array average (RMA) algorithm (Irizarry et al, 2003) available in the *affy* package (version 1.36.1) (Gautier et al, 2004). The log$_2$ scale data from RMA were used in statistical testing.

Differential expression analysis was performed, and the Benjamini–Hochberg false discovery rate (Benjamini & Hochberg, 1995) was implemented for both ANOVAs and *t* tests to generate corrected *P*-values (*q* values). A filtered gene list was generated for expression changes of greater than 2.0-fold and one-way ANOVA false discovery rate *q* < 0.01 and, furthermore, divided into four distinct clusters based on the expression pattern in the three conditions to generate a heatmap. Principal component analysis was performed by normalizing gene expression values across all samples to a mean of 0 and an SD of 1.0 with the *prcomp* R function. The three treatment conditions were separated with the first principal component (PC1), and replicates in each condition were separated with the second PC2.

## Pathway analyses

GO term enrichment analysis of clustered gene groups was performed using DAVID (Huang et al, 2009a, 2009b) with default settings. HomoloGene (version 68) (NCBI Resource Coordinators, 2013) was used to identify human homologs of the rat genes in the array. The R environment (version 3.4.3) was used for all microarray analyses. Kyoto Encyclopedia of Genes and Genomes (KEGG) pathway analysis was used to perform subsequent bioinformatics analysis of all the genes identified in microarray, irrespective of the clusters. Both GO term and KEGG analysis pathways were selected with a *P*-value < 0.05 and a gene count >2.

## Transcription factor and target network construction

Passing attributes between networks for data assimilation (PANDA) analysis was used to construct gene regulatory networks for each of three treatment drugs, Syk, Rapa, and DMSO, using all genes in the microarray dataset, as previously described (Glass et al, 2013). An initial map of transcription factors to genes was created by scanning the rn6 genome for 620 Cis-BP *Rattus norvegicus* motifs provided with the MEME suite (Bailey et al, 2009) using the Finding Motif Occurrences (FIMO) program (Grant et al, 2011). Statistically significant ($p < 1 \times 10^{-4}$) hits within the promoter region, defined as a the [−750,+250]-bp region around the transcriptional start site of RefSeq annotated genes, were retained. The initial mapping included 2,245,143 edges (17,177 genes, 616 TF). There were 14,890 genes and 616 TF common to both expression data and the motif mapping. Potential inferred regulatory relationships were determined by using PANDA to integrate the motif mapping and gene expression data and assign a weight (z-score) to each edge that connects a TF to its target gene. Top 10,000 (TF, gene) pairs with the largest absolute differences of edge weights in three different pairwise comparisons were generated: Pair 1. E(SykI)-E(DMSO), Pair 2. E(rapamycin)-E(DMSO), and Pair 3. E(rapamycin)-E(SykI). In addition, edge weight differences for Pair 3 were plotted as scatter plots for visualization using Cytoscape. Transcription factors involved in the top edges for each pairwise comparison were identified from the network of differential regulation.

## Real-time (RT)-qPCR

1 µg of total RNA extracted from ELT3-V cells using the RNeasy Mini Kit (QIAGEN) was reverse-transcribed into cDNA using amfiRivert cDNA Synthesis Master Mix (GenDEPOT). Quantitative real-time polymerase chain reaction (qPCR) was performed using iTaq Universal SYBR Green qPCR Master Mix (Bio-Rad), according to the manufacturer's protocol. Table S1 provides the primer sequences used in qPCR analyses.

## Cellular fractionation

Treated ELT3-V cells were washed with cold PBS and collected by scraping. Cellular fractionation was performed using a CelLytic NuCLEAR Extraction Kit (Sigma-Aldrich), according to the manufacturer's instructions. A portion of cell pellets was used for total protein isolation using RIPA buffer (Thermo Fisher Scientific). Both nuclear and total protein isolation buffers were supplemented with protease and phosphatase inhibitors (Invitrogen). Equal fractions of lysates for both nuclear and cytoplasmic fractions were subjected to immunoblotting.

## Immunoblot

Equal amounts or fractions of indicated protein lysates were loaded onto NuPage 4–12% Bis-Tris Protein Gels (Invitrogen) and then

control. Data are means ± SEM of at least three independent experiments (**$P$ < 0.01; *t* test). **(D, E)** Equal amounts of protein from whole-cell lysates of ELT3-V transfected with either Scr or Parpbp siRNA for 48 h were analyzed by Western blotting with antibodies against Parpbp and β-actin. The ratio of Parpbp to β-Actin density was expressed as the fold-change relative to Scr siRNA. Data are means ± SEM of at least three independent experiments (*$P$ < 0.05; *t* test). **(F, G, H, I)** Equal amounts of proteins were separated by electrophoresis and transferred to a polyvinylidene difluoride membrane which were reacted with antibodies against CHOP, pEIF-2α, total EIF-2α, cPARP, and uncleaved PARP. β-Actin was used as a loading control. **(G, H, I)** Ratios of CHOP (G), pEIF-2α/EIF-2α (H), and cPARP/PARP (I) to β-actin density was expressed as fold change to Scr siRNA. Data represent means ± SEM of at least three independent experiments (*$P$ < 0.05, *t* test).
Source data are available for this figure.

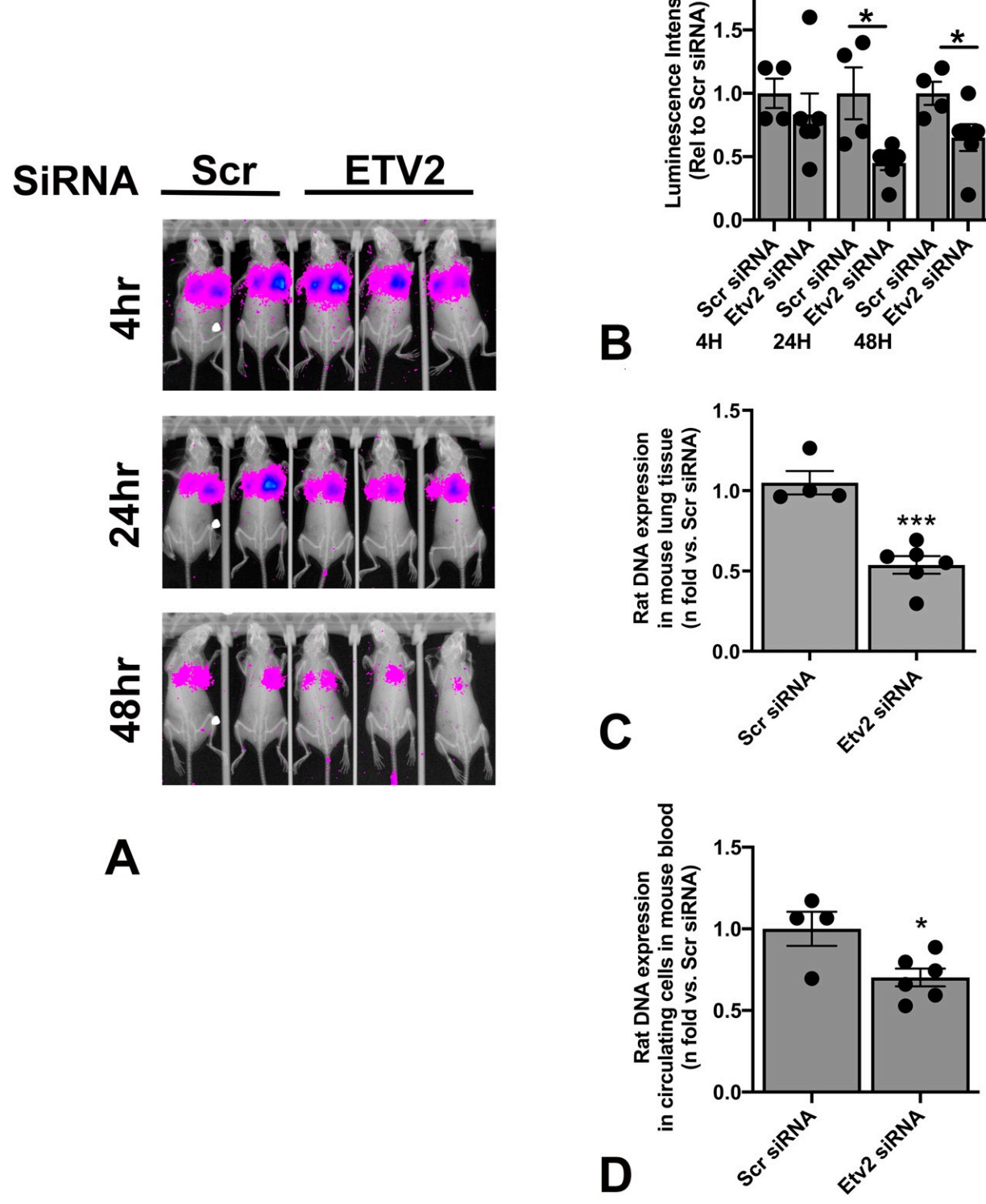

**Figure 5. Silencing *Etv2* leads to *Tsc2*-deficient cell death in vivo.**
**(A)** ELT3-V-luciferase cells were transfected with Scr and Etv2 siRNA for 24 h. 1 × 10$^6$ cells were injected in C.B17 Scid mice (Scr siRNA n = 4; Etv2 siRNA n = 6) via lateral tail vein injection. Bioluminescence was measured using Bruker In-Vivo Xtreme to show lung colonization of ELT3-V-luciferase cells at 4, 24, and 48 h. Representative images are presented. **(B)** Net luminescence intensity was accessed for each mouse at each time point. Histogram reveals a decrease in luminescence intensity in the Etv2 siRNA group relative to the Scr siRNA group at each time point. (*$P$ < 0.05; $t$ test). **(C)** 48-h postinjection, aortic blood was collected from each mouse. Levels of circulating cells from Scr siRNA (n = 4) and Etv2 siRNA (n = 6) mice were measured by RT-qPCR using rat-specific primers. The histogram represents fold change in rat DNA expression

subsequently immunoblotted with the primary antibodies listed in Table S2. Each protein of interest was then detected with HRP-conjugated goat anti-rabbit or anti-mouse IgG antibody (1:2,000; Invitrogen) and visualized using SuperSignal West Pico PLUS Chemiluminescent Substrate or SuperSignal West Femto Maximum Sensitivity Substrate (Thermo Fisher Scientific).

### RNA interference

Predesigned MISSION siRNA targeting rat *Etv2* and rat *Parpbp*, as well as MISSION siRNA Universal Negative control, were purchased from Sigma-Aldrich. ELT3-V cells were transfected with Etv2 siRNA (10 $\mu$M), Parpbp siRNA (5 $\mu$M), or scrambled (Scr) siRNA using Lipofectamine RNAiMAX reagent (Thermo Fisher Scientific) and OPTI-MEM (Thermo Fisher Scientific). Sequences for siRNA used are listed in Table S3.

### Arsenite-induced SG formation

ELT3-V cells were plated on four-well chamber tissue culture slides (Corning) and transfected with Scr or Etv2 siRNA. 48-h post-transfection, cells were treated with 0.5 mM sodium arsenite ($NaAsO_2$) (MilliporeSigma) for 40 min. The cells were then fixed with 4% paraformaldehyde and processed for immunofluorescence analysis using GAP SH3 Binding Protein 1 (G3BP1) antibody (green; Abcam) and fluorophore-conjugated secondary antibody. Nuclei were visualized with 4′,6-diamidino-2-phenylindole (DAPI, blue; Sigma-Aldrich) staining. Images were captured with a FluoView FV-10i Olympus Laser Point Scanning Confocal Microscope using a 60× objective (Olympus).

### Luciferase reporter assay

The *Parpbp* promoter fragments with (386 bp) or without (150 bp) one putative ETV2 binding site (Wei et al, 2010) were cloned into pGL3_Basic vector (Cat. no. E1751; Promega). Primers forward: taagcagagctcgtcggagggcgagcgaggcg and reverse: tgcttaaagcttttaccacgatgccgctggagg were used to clone −311 to +75 bp and primers forward: taagcagagctcggcgcggaacaagcgtagtagtcag and reverse: tgcttaaagcttttaccacgatgccgctggagg were used to clone −75 to +75 bp promoter fragments. Reverse transfection was carried out in a total of 15,000 ELT3-V cells transfected with Scr and Etv2 siRNA for 24 h using X-tremeGENE HP DNA transfection reagent (Sigma-Aldrich). Cells were transfected with 10 ng of pRL_CMV vector (internal control, Cat. no. E2261; Promega) and 140 ng of promoter constructs. Luciferase activity was tested 24 h after plasmid transfections using the Dual-Glo Luciferase Assay System kit (Cat. no. E2920; Promega) as per the manufacturer's protocol. Each transfection was carried out in triplicates in four independent experiments. Biotek Synergy HT microplate reader (BioTek) with Biotek Gen5.1.1 microplate data collection software was used for luciferase luminescence detection.

### Animal studies

All animal experimental procedures were performed according to protocols approved by the Institutional Animal Care and Use Committee at Brigham and Women's Hospital. ELT3-V cells stably transduced with pCMV-luciferase (ELT3-V-luciferase) were transfected with Scr siRNA and Etv2 siRNA. 24-h post-transfection, $1 \times 10^6$ cells were injected into female immunodeficient C.B17 SCID mice (Taconic) via lateral tail vein injections. Before imaging, mice were injected with the RediJect D-luciferin bioluminescent substrate (Cat. no. 770504; Perkin-Elmer). Bioluminescent signals were recorded at 4, 24, and 48 h using an In-Vivo Xtreme imaging system (Bruker). Net luminescence intensity in the chest area was assessed using Bruker molecular imaging (MI) software (v7). Murine blood was collected at the end of the experiment by aortic puncture, and red blood cells were lysed using an ammonium–chloride–potassium (ACK) lysing buffer (Quality Biological). In addition, murine lung tissues were harvested, minced, and enzymatically digested (300 units/ml Collagenase 4; Worthington Dorchester). DNA was extracted from blood and mouse lung tissues using a Blood and Tissue DNA kit (QIAGEN). Rat and mouse DNA were quantified by RT-qPCR using rat- and mouse-specific primers included in Table S1 (Walker et al, 2004; Yu et al, 2009).

### Analysis of available single-cell RNA-Seq data

scRNA seq-data for four LAM samples were downloaded from Gene Expression Omnibus (GSE135851) (Guo et al, 2020). Previous analysis of LAM sample 2 did not detect LAMCORE positive cells and therefore was excluded from our analysis. The data from LAM samples 1, 3, and 4 were analyzed using Seurat v3 (Butler et al, 2018). For the three LAM samples, genes expressed in at least two cells were selected. Furthermore, for LAM samples 1 and 3 specifically, cells with at least 500 expressed genes (nFeature_RNA > 500) with less than 10% of genes mapping to the mitochondria (percent.mt < 10) were selected. For LAM sample 4, nuclei with at least 300 expressed genes and with less than 10% of genes mapping to the mitochondria were included for analysis. The three Seurat objects were merged into a single object and preprocessed using the outline previously described (Butler et al, 2018). The samples were then integrated using the R package Harmony (Korsunsky et al, 2019), followed by imputation of the data using the R package Alra (Linderman et al, 2022). Clusters were then generated using Seurat and labeled with the R package ClustifyR (Fu et al, 2020) using previously published data as a reference matrix (Habermann et al, 2020). LAMCORE cells were identified based on positive (>0) expression of LAM markers *PMEL*, *FIGF*, *ACTA2*, and *VEGF-D* (Guo et al, 2020); 100 of these cells clustered together in the LAMCORE cluster. *ETV2*-positive cells within the LAMCORE cells were identified by positive (0>) expression of *ETV2*.

---

relative to Scr siRNA. **(D)** Rat DNA in the mouse lungs harvested 48-h postinjection from Scr siRNA (n = 4) and Etv2 siRNA (n = 6) mice were measured by RT-qPCR using rat-specific primers. The histogram represents fold change in rat DNA expression relative to Scr siRNA.

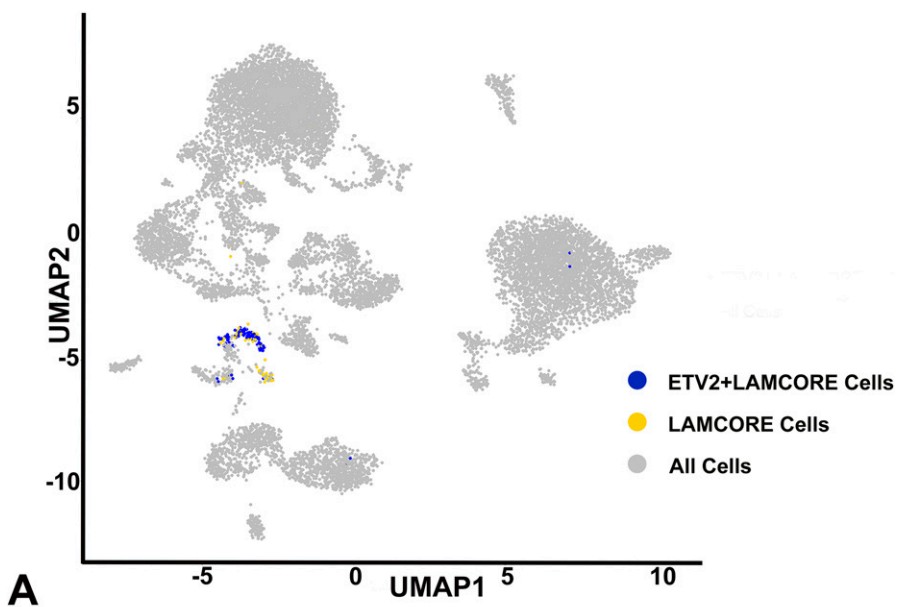

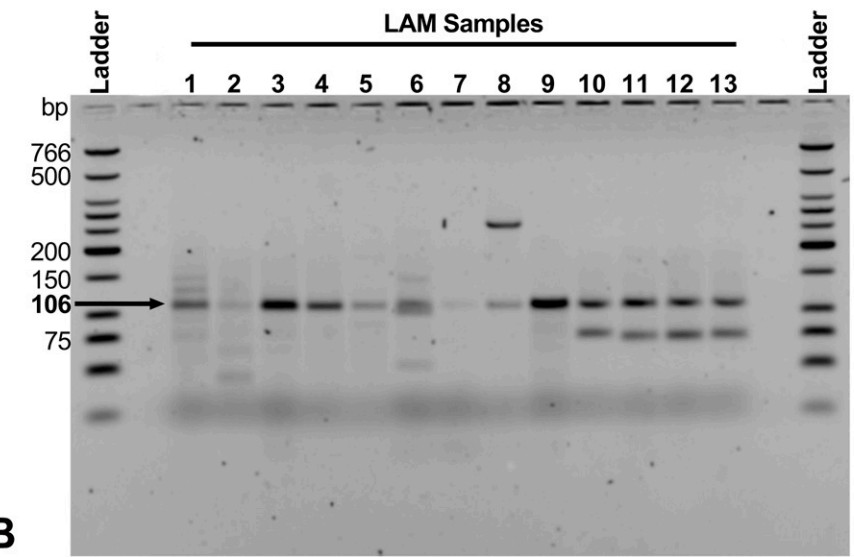

**Figure 6.  *ETV2* expression in human LAM samples.**
**(A)** Single cells RNAseq analysis of LAM samples detecting *ETV2* expressing cells in the LAMCORE population. Feature plot indicates 15 different clusters of cells (gray) and a distinct cluster of LAMCORE cells (yellow), in the LAM samples. *ETV2*-expressing cells within the LAMCORE cells are indicated in blue.
**(B)** Agarose gel electrophoresis analysis of *ETV2* expression in cultured LAM cells isolated from explanted lungs of patients undergoing lung transplantation (Samples 1–5) and circulating cells from human blood (Samples 6–13). Amplicons of 106 bp were obtained in all samples.
Source data are available for this figure.

## Isolation of cells with LOH for *TSC2*

Cells isolated from explanted lungs of patients undergoing lung transplantation were cultured in mesenchymal stem cell media (MSCGM) as described previously (Pacheco-Rodriguez et al, 2007). Cultured cells (~1 × 10⁶) were reacted with 20 $\mu$l antibodies of each anti-CD44v6-FITC and anti-CD44-PE (Table S2), following trypsinization. Cells underwent a 30-min incubation at room temperature. Then, 3 ml of PBS was added, and the tubes were centrifuged for 10 min at ~250$g$. The cell pellet was then suspended in 500 $\mu$l PBS. Cell mixtures were placed on ice until sorted using a BD FACSaria 2 (Becton Dickinson). Four subpopulations of cells were obtained in PBS and RNA later (Sigma-Aldrich). We previously showed that cells sorted expressing *CD44* and *CD44v6* were more likely to have LOH for *TSC2*. To identify circulating LAM cells, we took 60 ml of heparinized fresh blood to isolate cells, following a density gradient. For

that purpose, we used CD235a-PE and CD45-FITC antibodies (Table S2). Cells with *TSC2* LOH are most likely to be found in cells expressing *CD235a*. *TSC2* LOH was determined using five microsatellite markers (D16S521, D16S3024, D16S3395, Kg8, and D16S291) on chromosome 16, as previously described (Steagall et al, 2013).

Total RNA was isolated from cells in RNALater, cDNA synthesized, and qPCR performed, according to the manufacturer's protocol. Primer sequences used in qPCR analyses were provided in Table S1. PCR products were subjected to agarose gel (2%) electrophoresis to visualize the amplified *ETV2* qPCR products.

## Statistical analysis

For all experiments, at least three independent experiments were conducted, and data are presented as mean ± SEM. Statistical

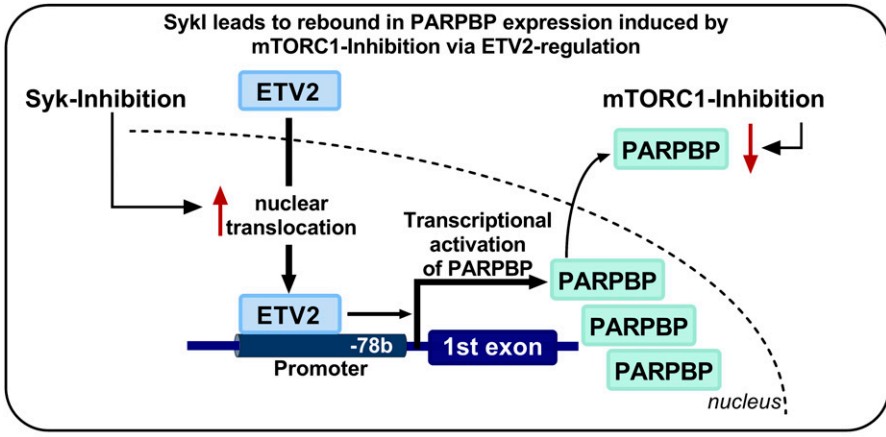

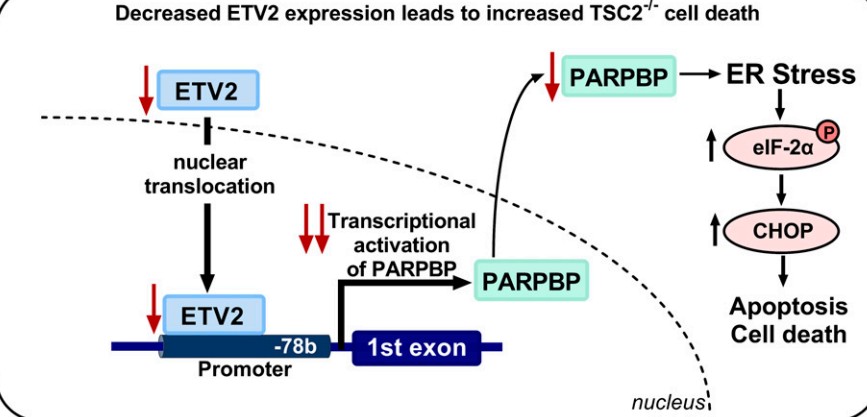

**Figure 7. Schematic representation of ETV2 nuclear translocation and PARPBP regulation.**
(Top panel) In *Tsc2*-deficient cells, Syk inhibition (SykI) drives ETV2 to the nucleus, where it binds to its consensus sequence in the *Parpbp* promoter at position −78 base pair to transcriptionally activation of *Parpbp* gene expression. (Bottom panel) In *Tsc2*-deficient cells, decreased ETV2 expression results in reduced *Parpbp* mRNA and PARPBP protein. Reduction in ETV2 or PARPBP protein levels induces ER stress, with increased phosphorylation of eIF-2a and CHOP expression, resulting in increased apoptosis and cell death.

analyses of all endpoints were performed using one-way ANOVA, followed by a Tukey post hoc test or one- or two-tailed *t* test. $P < 0.05$ was considered statistically significant. Analyses were performed using GraphPad Prism 8.3 (GraphPad Software).

# Data Availability

The microarray data from this publication have been deposited to the Gene Expression Omnibus database and assigned the identifier ID GSE183110 on the following link: https://www.ncbi.nlm.nih.gov/geo/query/acc.cgi?acc=GSE183110.

# Supplementary Information

# Acknowledgements

We are indebted to patients who participated in our protocols. This work was supported in part by the National Institutes of Health (U01-HL 131022 to S El-Chemaly and T32HL007633-35 to J Ng) and the Division of Intramural Research, National Institutes of Health, National Heart, Lung, and Blood Institute (J Moss) and the Anne Levine LAM Research Fund (S El-Chemaly).

## Author Contributions

S Shrestha: conceptualization, data curation, formal analysis, investigation, methodology, and writing—original draft, review, and editing.
A Lamattina: data curation, formal analysis, investigation, and writing—review and editing.
G Pacheco-Rodriguez: conceptualization, data curation, investigation, and writing—review and editing.
J Ng: conceptualization, formal analysis, investigation, and writing—review and editing.
X Liu: data curation, investigation, and writing—review and editing.
A Sonawane: formal analysis, investigation, methodology, and writing—review and editing.
J Imani: data curation, formal analysis, investigation, and writing—review and editing.
W Qiu: formal analysis, investigation, methodology, and writing—review and editing.
K Kosmas: data curation, formal analysis, investigation, and writing—review and editing.
P Louis: formal analysis, investigation, and writing—review and editing.

A Hentschel: investigation and writing—review and editing.

WK Steagall: conceptualization, data curation, formal analysis, investigation, and writing—review and editing.

R Onishi: data curation, formal analysis, and investigation.

H Christou: conceptualization, formal analysis, and writing—review and editing.

EP Henske: formal analysis, methodology, and writing—review and editing.

K Glass: conceptualization, formal analysis, investigation, methodology, and writing—review and editing.

MA Perrella: conceptualization, formal analysis, investigation, methodology, and writing—review and editing.

J Moss: conceptualization, formal analysis, supervision, funding acquisition, investigation, project administration, and writing—review and editing.

K Tantisira: conceptualization, formal analysis, methodology, and writing—review and editing.

S El-Chemaly: conceptualization, data curation, formal analysis, supervision, funding acquisition, investigation, methodology, project administration, and writing—original draft, review, and editing.

## Conflict of Interest Statement

The authors declare that they have no conflict of interest.

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
