## [Reviewer comments · Life Science Alliance]

ETV2 regulates PARP-1 binding protein to induce ER stress-mediated death in tuberin-deficient cells

Shikshya Shrestha, Anthony Lamattina, Gustavo Pacheco-Rodriguez, Julie Ng, Xiaoli Liu, Abhijeet Sonawane, Jewel Imani, Weiliang Qiu, Kosmas Kosmas, Pierce Louis, Anne Hentschel, Wendy Steagall, Rieko Onishi, Helen Christou, Elizabeth Henske, Kimberly Glass, Mark Perrella, Joel Moss, Kelan Tantisira, and Souheil El-Chemaly

DOI: <https://doi.org/10.26508/lsa.202201369>

Corresponding author(s): *Souheil El-Chemaly, Brigham and Women's Hospital and Shikshya Shrestha, Brigham and Women's Hospital*

Review Timeline:	Submission Date:	2022-01-10
	Editorial Decision:	2022-01-19
	Revision Received:	2022-01-21
	Accepted:	2022-01-21

Scientific Editor: *Eric Sawey, PhD*

Transaction Report:

Please note that the manuscript was previously reviewed at another journal and the reports were taken into account in the decision-making process at *Life Science Alliance*. Since the original reviews are not subject to Life Science Alliance's transparent review process policy, the reports and author response cannot be published.

January 19, 2022

RE: Life Science Alliance Manuscript #LSA-2022-01369

Dr. Souheil El-Chemaly
Brigham and Women's Hospital
75 Francis Street
Boston 02446

Dear Dr. El-Chemaly,

Thank you for submitting your revised manuscript entitled "ETV2 regulates PARP-1 binding protein to induce ER stress-mediated death in tuberin-deficient cells". We would be happy to publish your paper in Life Science Alliance pending final revisions necessary to meet our formatting guidelines.

- please upload your main manuscript text as an editable doc file
- please upload your main and supplementary figures as single files
- please add the Twitter handle of your host institute/organization as well as your own or/and one of the authors in our system
- please upload your Tables in editable .doc or excel format
- please note that titles in the system and manuscript file must match
- please add your main, supplementary figure, and table legends to the main manuscript text after the references section
- please consult our manuscript preparation guidelines <https://www.life-science-alliance.org/manuscript-prep> and make sure your manuscript sections are in the correct order
- please use the [10 author names, et al.] format in your references (i.e. limit the author names to the first 10)
- please add callouts for Figures 7A-B; S2A-G; S3A-D and S4A-D to your main manuscript text

FIGURE CHECKS:

- the minimum resolution for all figures should be at least 300 dpi
- the visibility of scale bars in figure 3J should be increased

A. FINAL FILES:

B. MANUSCRIPT ORGANIZATION AND FORMATTING:

Sincerely,

January 21, 2022

RE: Life Science Alliance Manuscript #LSA-2022-01369R

Dr. Souheil El-Chemaly
Brigham and Women's Hospital
75 Francis Street
Boston 02446

Dear Dr. El-Chemaly,

Thank you for submitting your Research Article entitled "ETV2 regulates PARP-1 binding protein to induce ER stress-mediated death in tuberin-deficient cells". It is a pleasure to let you know that your manuscript is now accepted for publication in Life Science Alliance. Congratulations on this interesting work.

DISTRIBUTION OF MATERIALS:

Again, congratulations on a very nice paper. I hope you found the review process to be constructive and are pleased with how the manuscript was handled editorially. We look forward to future exciting submissions from your lab.

Sincerely,
